# Hardening Properties of Cheeses by *Latilactobacillus curvatus* PD1 Isolated from Hardened Cheese-*Dd**ukbokki* Rice Cake

**DOI:** 10.3390/microorganisms9051044

**Published:** 2021-05-12

**Authors:** Jeong-A. Kim, Geun-Su Kim, Se-Mi Choi, Myeong-Seon Kim, Do-Young Kwon, Sang-Gu Kim, Sang-Yun Lee, Kang-Wook Lee

**Affiliations:** Pulmuone Institute of Technology, Cheongju 28164, Korea; crystal.kim01@pulmuone.com (J.-A.K.); gskimb@pulmuone.com (G.-S.K.); smchoi@pulmuone.com (S.-M.C.); sunny.kim01@pulmuone.com (M.-S.K.); dykwon@pulmuone.com (D.-Y.K.); sgkimn@pulmuone.com (S.-G.K.); sylee@pulmuone.com (S.-Y.L.)

**Keywords:** cheese, cheese hardening, *Latilactobacillus curvatus*, cheese-*ddukbokki* rice cake

## Abstract

Hardening of cheese is one of major issues that degrade the quality of Home Meal Replacement (HMR) foods containing cheese such as Cheese-*ddukbokki* rice cake (CD, stir-fried rice cakes with shredded cheese). The quality of cheese, such as pH, proteolytic, and flavor properties, depends on various lactic acid bacteria (LAB) used in cheese fermentation. The hardening of cheese is also caused by LAB. In this study, various LAB strains were isolated from CD samples that showed rapid hardening. The correlation of LAB with the hardening of cheese was investigated. Seven of the CD samples with different manufacturing dates were collected and tested for hardening properties of cheese. Among them, strong-hardening of cheese was confirmed for two samples and weak-hardening was confirmed for one sample. All LAB in two strong-hardening samples and 40% of LAB in one weak-hardening sample were identified as *Latilactobacillus*
*curvatus*. On the other hand, most LAB in normal cheese samples were identified as *Leuconostoc mesenteroides* and *Lactobacillus casei*. We prepared cheese samples in which *L. curvatus* (LC-CD) and *L. mesenteroides* (LM-CD) were most dominant, respectively. Each CD made of the prepared cheese was subjected to quality test for 50 days at 10 °C. Hardening of cheese with LC-CD dominant appeared at 30 days. However, hardening of cheese with LM-CD dominant did not appear until 50 days. The pH of the LC-CD was 5.18 ± 0.04 at 30 days, lower than that of LM-CD. The proteolytic activity of LC-CD sample was 2993.67 ± 246.17 units/g, higher than that of LM-CD sample (1421.67 ± 174.5 units/g). These results indicate that high acid production and high protease activity of *L. curvatus* might have caused hardening of cheese.

## 1. Introduction

Cheese is a fermented dairy product made by coagulating casein, removing whey, compressing and molding by adding lactic acid bacteria (LAB), rennet, and acid to milk or cream. Cheeses are manufactured through several steps, such as gel-forming, whey expulsion, salt addition, and acid production [1,2]. The quality of cheese differs depending on the raw milk, microorganisms, enzymes, and processes used for manufacturing, and the manufacturing region. It has been reported that approximately 1000 kinds of cheese or more are produced worldwide [2]. Types of cheese are generally classified according to characteristics of the cheese. Cheese can be classified into soft, semi-soft, semi-hard, and hard cheese depending on its moisture content and the texture. Cheese typically shows three forms when heated, including a stretchy form with a property that stretches like rubber, a creamy form, and a non-melting form that crumbles [3].

In Korea, according to statistical results of types of cheese purchased by consumers in 2015, shredded cheese is widely consumed as one of the preferred cheese types for all age groups [4,5]. However, shredded cheese has a larger surface area than other types of cheese. Its quality is relatively unstable due to rapid moisture reduction, rancidification, and over-fermentation by microorganisms. To maintain the quality of cheese, adding preservatives and filling gas are performed during cheese processing and packaging stages. For shredded cheese that is usually stored at 0–10 °C, a mixture of nitrogen and carbon dioxide is filled inside the package [6]. Cellulose powder is generally used as an anti-caking agent during the shredding process. [3]. Shredded cheese is widely used in home meal replacement foods (HMR). Korean traditional types of HMR foods such as *ddukbokki* (spicy stir-fried rice cakes), *kimchi-jeon* (Kimchi pancake), and cheese ball (fried cheese bread) are very popular. They are exported to other countries such as United States, Japan, and China. With increasing demand for HMR foods with cheese, there are also many manufacturing methods. Therefore, types of final food products have become very diverse depending on the processing method, the storage method, and the type of raw materials. It is very important to maintain the best quality of each product.

*Ddukbbokki* is one of Korea’s traditional types of HMR foods. It is prepared with *dduk* (rice cake) and seasoning, like soy sauce, sugar, and *kochujang*. *Ddukbokki-dduk (Garae-dduk),* the major material of *ddukbokki*, is one of various rice cakes in Korea. *Garae-dduk* is manufactured by kneading rice or flour in water, forming into a rod shape, and steaming at a high temperature [7].

Research studies on the quality of cheese are continuously reported. Studies on starters such as *Streptococcus thermophiles**, Leuconostoc* sp., *Lactococcus* sp., *Lactobacillus* sp., *Bifidobacterium* sp., and *Penicillium* sp. used in cheese are representatives [1]. The major role of LAB in cheese fermentation is that LAB can decrease pH values by producing various organic acids. The low pH of cheese affects its texture and rheological properties directly related to chemical changes in the protein network of cheese curd [8]. Some LAB have proteolytic activity and can hydrolyze oligopeptides into smaller peptides and amino acids [9,10,11,12]. Each starter of cheese has individual properties, such as proteolytic activity and acid production. These starters can affect the properties of the final cheese product. Among them, proteolytic activity is a good indicator for ripening and the development of cheese flavor [13,14,15], but presumably a negative indicator for over ripening of cheese products. In non-melting cheese, such as cottage cheese, protease activity is used as a positive indicator [16].

*Latilactobacillus curvatus* is one of the generally-detected typical facultative hetero-fermentative LAB from fermented foods such as cheese [17,18]. Mikelsone et al. have reported that it is mainly detected in Krievijas cheese until ripening at 60 days [19]. There have been various reports on *L. curvatus* that can be referred to in this study. *L. curvatus* produces more amounts of organic acid than other LAB [20]. It can survive at cold chain temperature for 30 days or more on dairy products [21]. In addition, some studies have reported that *L. curvatus* isolated from cheese possesses antimicrobial activities [22,23].

Previously, many studies on cheese, such as cheese starter, flavor, and texture, have been reported [1]. However, negative effects of specific microorganism on hardening of cheese have not been reported yet. In this study, *L. curvatus* was isolated from cheese samples that showed hardening and its correlation with the hardening of cheese was investigated.

## 2. Materials and Methods

### 2.1. Isolation and Identification of Lactic Acid Bacteria Strains from Cheese and Cheese-Ddukbokki Rice Cake

*Ddukbokki* rice cake is one of the Korean traditional convenience foods, is made by soaking and kneading rice or wheat with water, steaming and shape at high temperature above 90 °C. Steaming process can kill other bacteria, including LAB, derived from grains such as rice and wheat. In Korea, shredded cheese is often added into steamed rice cake, it is called cheese-*ddukbokki* rice cake (CD). Most CD is commercialized to a cylinder shape of rice cake surrounding the shredded cheese in Korea (Figure 1).

Pizza Shredded Ⅱ mix-5 cheese (PS5, Mozzarella cheese: Gouda cheese = mixture of 4:1, material of CD samples, Maeil Dairies Co. Ltd., Gochang, Jeonbuk, Korea) was purchased at a cheese process company and a department store in Cheongju, Chungbuk, Korea from September to December of 2020. Seven cheese-*ddukbokki* rice cake (CD, Pulmuone, Goesan, Chungbuk, Korea) CD samples with different manufacturing dates were collected and used.

Twenty-five grams of CD and PS5 samples were mixed with 225 mL 0.85% NaCl solution and homogenized using a stomacher (BagMixer 400, Interscience, Saint-Nom-la-Bretèche, France). Homogenates were diluted serially and spread onto MRS agar medium (BD, Difco, Sparks, Maryland, USA). The MRS plates were then incubated at 35 °C for 72 h and 30 colonies were selected randomly according to shape, and size. Selected colonies were identified by molecular biological method. 16S rRNA genes of isolates were amplified by PCR. Emerald-Amp PCR Master mix (TAKARA, Kusatsu, Shiga, Japan) was used with primer pairs: 27F (5′-AGAGTTTGATCMTGGCTCAG-3′) and 1492R (5′-TACGGYTACCTTGTTACGACTT-3′) [24]. Thermal Cycler Dice Touch (TAKARA, Kusatsu, Shiga, Japan) was used for PCR. Sequences were determined at MACROGEN (Seoul, Korea) and analyzed using BLAST (NCBI, Bethesda, MD, USA).

### 2.2. Growth Properties of Latilactobacillus curvatus at Different Temperature Conditions

*L. curvatus* isolate was initially grown in MRS broth at 35 °C for 72 h. The bacterial cells were inoculated (1%, *v/v*) to fresh MRS broth with 2% (*w/v*) lactose (DUKSAN pure chemical, Ansan, Gyeonggi, Korea). These cultures were cultivated at different temperatures (5, 10, 15, 25 and 35 °C) for 6 days. The growth was monitored by measuring the OD_600_ values by micro-plate reader (infinite 200 pro, TECAN, Männedorf, Switzerland) at 5, 9 h, then 24 h intervals and pH value was monitored with a Potentionmetric Titrator (SI Analytics, Mainz, Germany) every 24 h.

### 2.3. Preparation of Cheese-Ddukbokki Rice Cake Containing Different Dominant Lactic Acid Bacteria

Two types of *cheese-ddukbokki* rice cake containing *L. curvatus* PD1 or *L.*
*mesenteroides* PD2 were prepared at Pulmuone R&D center (Pulmuone Institute of Technology, Cheongju, Chungbuk, Korea). CD samples were manufactured by kneading and mixing rice flour with water and steaming at high temperature. Afterward shredded cheese was added to the steamed dough, it was re-molded in a short cylindroid shape. Cheese was added at 10% (*w/w*) based on the weight of cheese-*ddukbokki* rice cake using two types of cheese samples in which *L. curvatus* (LC-CD) or *L.*
*mesenteroides* (LM-CD) was dominant. Commercially-available CD samples containing cheese without dominant starter were purchased as controls and used in this study. Each CD made of prepared cheese was subjected to quality test at 10 °C for 50 days.

### 2.4. pH, Water Content and Meltability Test of Cheese-Ddukbokki Rice Cake

Twenty-five grams of CD sample was mixed with 225 mL of distilled water and homogenized using BagMixer 400 (Interscience, Saint-Nom-la-Bretèche, France) with a filter bag (Whirl-Pak, Nasco, Madison, USA). The pH of the filtered sample was measured using a Potentionmetric Titrator (SI Analytics, Mainz, Germany). Water content of CD sample was measured for 50 days according to loss after drying by Korea Food Code [25]. Five grams of cheese was separated from CD sample and dried at 105 °C for 5 h in a dry oven. Drying samples were cooled down at room temperature for 30 min. Weights of samples were measured and water content (%) was calculated.

Meltability test was performed for CD samples using a water bath (WiseBath, DAIHAN, Wonju, Gangwon, Korea) every 10 days during storage. Twenty grams of each CD and five grams of cheese separated from CD samples were used for the test. CD and cheese samples were placed in a conical tube (30 × 115 mm, SPL, Pocheon, Gyeonggi, Korea) and wrapped with aluminum-foil. These conical tubes with sample were heated at 80 °C for 30 min. The melting property of cheese was determined according to the appearance of cheese (melting or non-melting).

### 2.5. Microbial Analysis of Cheese-Ddukbokki Rice Cake

Microbial analysis was performed for CD sample every 10 days for 50 days. Twenty-five grams of CD was mixed with 225 mL of 0.85% NaCl solution (*w/v*) and homogenized using a stomacher (BagMixer 400, Interscience, Saint-Nom-la-Bretèche, France). Homogenized sample was diluted serially and spread onto MRS agar plates. MRS plates were incubated at 35 °C for 72 h and 40 colonies were selected randomly according to shape, size, and color. Selected colonies were identified by molecular biological method as described in Materials and Methods Section 2.3.

### 2.6. Proteolytic Activity of Lactic Acid Bacteria and Cheese-Ddukbokki Rice Cake

Proteolytic activities of CD samples were measured according to methods by Korea Food code. Five grams of CD sample was mixed with 95 mL distilled water. Supernatant was obtained after centrifugation (12,000 rpm, 10 min, 4 °C). Then 1 mL of supernatant sample and casein (Sigma-Aldrich, Saint Louis, MO, USA) solution (0.6%, *w/v*) were mixed well and incubated at 37 °C for 10 min. Then 2 mL of 0.4M TCA solution was added and incubated at 37 °C for 25 min. Samples were filtered using a syringe filter (0.45 μm, Pall Medical, New York, NY, USA). One ml of filtered sample was mixed with 5 mL of Na_2_CO_3_ and 1 mL of folin solution (3-fold dilution with distilled water, Sigma-Aldrich, Saint Louis, MO, USA). The absorbance of sample was measured at 660 nm after incubating at 37 °C for 20 min. A standard curve was prepared using tyrosine at different concentrations. One unit of enzyme activity was expressed as the amount of enzyme that released 1 μmol of tyrosine per min. Proteolytic activities of LAB isolates from shredded cheese were measured according to sigma’s non-specific activity method as described previously [26].

## 3. Results and Discussion

### 3.1. Isolation and Identification of Lactic Acid Bacteria Strains

Fifteen of the PS5 samples and seven CD samples were collected to check whether hardening occurred during the storage period and LAB were isolated (Table 1 and Table 2). Among them, strong-hardening of cheese was found for two samples and weak-hardening was found for one sample. In addition, sour-taste was observed for cheese samples showing hardening. All LAB isolates from strong-hardening samples and 40% of LAB isolates from weak-hardening sample were identified as *Lactobacillus curvatus* (Gene bank number of *L. curvatus* PD1 in this study: MW750572). On the other hand, most LAB isolates from normal cheese samples were identified as *Leuconostoc*
*mesenteroides* (Gene bank number of *L.*
*mesenteroides* PD2 in this study: MW750574) and *Lactobacillus casei*. All LAB strain’s identity percentages were 99.5–100% in BLAST program. As the dominant strain in the hardening-cheese was identified as *L. curvatus*, we determined that this strain was an indirect cause of hardening of cheese.

There are many reports of LAB used as cheese starters, such as *Leuconostoc* sp., *Lactococcus* sp., and *Lactobacillus* sp. Among them, *L. curvatus* was commonly detected from initial to 60 days of Krievijas cheese ripening. In commercial Latvian cheese samples (non-starter cheese), *L. curvatus* (28.6%), *L. paracasei* subsp. *paracasei* (38%), *L. plantarum* (14.3%), *L. rhamnosus* (14.3%), and *L. acidophilus* (4.8%) were detected [27]. *L.*
*mesenteroides* was the dominant acidifying strain used. *L. lactis* was used to improve the flavor during fermentation of dairy product.

### 3.2. Growth Properties of Latilactobacillus curvatus at Different Temperature

*L. curvatus* isolated from hardening cheese-*ddukbokki* rice cake was incubated at 5, 10, 15, 25, and 35 °C for 7 days. OD_600_ values for growth monitoring and pH values were then measured at 5, 9 h, then every 24 h (Figure 2 and Figure 3). *L. mesenteroide**s* isolated from normal cheese-*ddukbokki* rice cake (non-hardening samples) was used as control. *L. curvatus* grew well at all temperature for 7 days (Figure 2A). At 15 to 35 °C, *L. curvatus* was better than 10 and 5 °C. *L. curvatus* more grew well at 25 and 15 °C, it showed that *L. curvatus* is one of the psychrotrophic bacteria. OD_600_ of *L. curvatus* cultures grown at 15–25 °C were the highest at 1.2674–1.3662 on day 4. At 10 °C, *L. curvatus* was increased steadily, showing the highest OD_600_ value at 0.6240 ± 0.02 on day 7. Its growth at 5 °C was slower (0.1538 ± 0.01 at 7 days) than under other conditions. *L.*
*mesenteroides* isolated from normal CD samples grew well to 1.2289–1.2710 at 25–35 °C after 7 days (Figure 2B). However, OD_600_ at 10 °C were lower than those for *L. curvatus* (OD_600_, 0.2169 ± 0.03). At 5 °C, the strain did not grow at OD_600_ values (0.1234 ± 0.01).

The initial pH value of *L. curvatus* culture was 6.38 ± 0.11. It decreased at all temperature conditions for 7 days (Figure 3A). At 5 °C, the pH value decreased slowly until day 7 (5.89 ± 0.01). It decreased to 5.08 ± 0.01 at 10 °C. At 15–25 °C, the lowest pH value was 3.94–4.16 at day 7. The pH value of *L. mesensteroides* culture was higher than that of *L. curvatus* at all temperature conditions (Figure 3B). The lowest pH value was 6.15 ± 0.06 for 5 °C, 5.85 ± 0.07 for 10 °C, 5.11 ± 0.13 for 15 °C, 4.39 ± 0.01 for 25 °C, and 4.46 ± 0.06 for 35 °C at 7 days.

LAB strains usually produce lactate during growth with decreased pH value. It is expected that the increase of LAB counts is related to the decrease of pH value during storage. Porcellato et al. have reported that *L. curvatus* isolated from cheddar cheese produces various organic acids such as lactate, citrate, and acetate [17]. The genus *Lactobacillus* is known as one of LAB with relatively strong acid resistance and acid-producing ability compared to other genera. *When Suan Cai* (pickled Chinese cabbage) fermentation in Northeast China used *L. curvatus* as a starter, pH value decreased more rapidly than when *L. mesenteroides* was used as starter [20]. It is expected that *L. curvatus* will be dominant in CD sample during cold storage. It has been reported that *L. cruvatus* DK-13 isolated from watery Kimchi can produce high amounts of lactic acid and acetic acid during growth on MRS broth [28]. A starter of sausage, *L. curvatus* 54M16 can grow at a minimum pH of 3.5 at 30 °C. This strain has a strong acid production [29].

### 3.3. pH, Water Content, and Meltability Test of Cheese-Ddukbokki Rice Cake

pH and water content (WC, %) of cheese-*ddukbokki* rice cake samples were measured every 10 days until 50 days (Figure 4). The pH value of CD sample (control) gradually decreased to 5.20 ± 0.11 at day 50 (Figure 4A). Initial pH values of LC-CD and LM-CD samples were 5.67 ± 0.09 and 5.77 ± 0.01, respectively (Figure 4B,C). Initial water content of LC-CD and LM-CD samples were 46.52 ± 0.50% (*w/w*) and 47.71 ± 0.15%(*w/w*), respectively. The pH value and water content of LC-CD sample were decreased to 5.18 ± 0.04 and 40.05 ± 1.63% (*w/w*), respectively, at day 30. They were then decreased to 5.09 ± 0.05 and 37.35 ± 0.21% (*w/w*), respectively, at day 50. At 50 days, the LC-CD cheese sample did not melt in the meltability test. The pH value of cheese is an important indicator of the quality of cheese. It is affected by chemical interactions and structural components of cheese, such as moisture, protein, and minerals [8]. The pH value of cheese affects its structure and rheological properties because chemical interactions between major structural components of protein are strongly pH dependent. In the case of semi-rigid mozzarella and gouda shred cheese, it has been reported that these cheeses do not melt. They can easily break without being stretched when the pH value is less than 5.0 [30].

The quality of semi-hard and hard cheese is generally suitable when the pH is 5.0 to 6.0. When the pH is weakly acidic which increases the anionic value of the casein protein to deform the matrix, thereby improving the water holding capacity and emulsifying ability. Cheeses with pH below the range of 5.0–6.0 are crumbly because low pH can affect the protein structure, calcium phosphate, and water content. Therefore, a reduced water content of cheese affects the hardness of cheese product [31]. In this study, the used PS5 cheese is a shredded product and mixed with mozzarella and gouda, so it is suitable in similar pH value within the same range (pH 5.0–6.0) as semi-hard and hard cheeses.

The meltability tests of CD samples at day 0 to day 50 were performed by boiling in hot water for 5 min (Table 3). Cheese of Con-CD was not melted. It had a strong sour taste at 50 days. The pH value of Con-CD sample was 5.18 ± 0.15 at 50 days. The meltability was determined for LC-CD and LM-CD samples at an interval of 10 days until 50 days (Table 2). At 30 days, LC-CD samples were not melting. They showed crumbling (Figure 5A,C). However, LM-CD samples were melting until day 50 (Figure 5B,D). The pH value of LC-CD sample at day 30 was 5.18 ± 0.04. It decreased gradually until day 50. Con-CD and LC-CD cheese samples did not melt. They were harder than other samples. Their pH values were less than 5.2. Lawrence et al. [8]. have reported that the stretchability and cohesion of natural cheese curd depend on the pH value. At pH below 5.2, small casein aggregates can bind together in long chains when heat is applied. When pH is at about pH 4.8, cohesion and stretchability of cheese are lost. Hardening properties and pH changes of CD and LC-CD samples showed similar trends to those reported by Lawrence et.al. [8]

### 3.4. Microbial Analysis of Cheese-Ddukbokki Rice Cake

Total LAB counts were determined and microbial community analyses of Con-CD samples were performed every 10 days. Results are shown in Figure 6. The initial LAB cell count of CD sample was 8.15 log CFU/g. It then increased to a count of 9.72 log CFU/g at 50 days. Forty colonies of LAB on MRS agar plates were randomly sequenced. Sequences were analyzed by BLAST for identification. *L.*
*mesenteroides* was the most dominant group in CD sample at day 0, accounting for 77.50% (31 out of 40). *L.*
*mesenteroides*, a heterofermentative bacterium, is often used as a commercial cheese starter [12]. *L. curvatus* was the next dominant group in CD samples at day 0 (22.50%, 9 out of 40). However, as cheese storage time at 10 to 50 days, the genus *Lactobacillus* becomes the dominant species. At 10 days, *L. paracasei* was the most dominant group (67.6%, 27 out of 40) and *L. curvatus* appeared as the most dominant species after 20 days of storage time. In particular, after 40 days, all strains were identified as *L. curvatus*.

Microbial community analysis results of LC-CD and LM-CD samples at 0 day and 30 days are shown in Table 4. In initial analysis, *L. curvatus* and *L.*
*mesenteroides* were the most dominant in LC-CD (100%) and LM-CD sample (100%), respectively. After 30 days, *L. curvatus* was still the dominant strain in LC-CD samples, whereas in LM-CD samples, *L.*
*mesenteroides* and *L. curvatus* appeared at a ratio of 8:2. The hardening of cheese was confirmed only for LC-CD cheese samples. These results suggest that *L. curvatus* is the cause of hardening of cheese more than other LAB strains such as *Leuconostoc* sp. and *Lactococcus* sp.

*L. curvatus* is considered as one of psychrotophic strains. A psychrotrophic strain is defined as one that can grow at 7 °C, although its optimal growth temperature is higher than 7 °C [32]. During cold storage after milk collection, they dominate the flora. Their extracellular enzymes, mainly proteases, contribute to the spoilage of dairy products. Especially, souring flavor, gas formation, slime production, and decreased pH are caused by bacteria in ready-to-eat like salads, raw fish, cheese, and fruits [17]. It has been reported that *L. curvatus* can produce slits in cheddar cheese, cause spoilage of processed modified-atmosphere-packaged (MAP) meat products, and produce biogenic amine in cold-smoked salmon and Dutch-type cheese [33,34,35].

### 3.5. Proteolytic Activity of Lactic Acid Bacteria and Cheese-Ddukbokki Rice Cake

Proteolytic activity of *cheese-ddukbokki* rice cake is shown in Figure 7. The proteolytic activity of Con-CD sample was 1296 ± 50.45 units/g at day 0. It gradually increased to 6170 ± 144.40 units/g at day 50. The proteolytic activity of LC-CD cheese sample increased rapidly to 4944.67 ± 167.55 units/g until day 30. Its proteolytic activity was higher than that of Con-CD or LM-CD cheese sample. However, at day 50 of storage, it showed a proteolytic activity similar to Con-CD sample. Proteolytic activity of LM-CD was the lowest at all storage time. Six LAB strains (*L. curvatus, L. paracasei, L. casei, L. mesenteroide*, *L. pesudomesenteroid**e,* and *L. sakei*) isolated from shredded cheese and Con-CD sample were measured for their proteolytic activities (Figure 7B). The proteolytic activity of *Lactobacillus* species was higher than that of other species, with *L. curvatus* having the highest activity at 135 ± 18.38 Unit/mL, followed by *L.paracasei* (94.5 ± 4.95 Unit/mL) and *L.*
*mesenteroides* (12.5 ± 7.78 Unit/mL). Generally, protease activity of LAB strain in cheese is a positive indicator of improved flavor during ripening. However, for melting-cheese such as shredded mozzarella cheese, protease activity is a negative indicator that must be carefully controlled.

Proteolysis is an essential process for the growth of LAB in milk and dairy products. This proteolysis provides essential free amino acids for the growth of LAB whose biosynthesis for some amino acids is limited. In addition, vitamins and minerals are produced during this biosynthesis process. Previously studies have reported that *Lactobacillus* sp. can produce vitamins [3,36,37]. Most LAB can produce a cell envelope proteinase essential for their optimal growth. Hydrolysis of cheese casein by cell envelope proteinase is considered to be the first and essential step in nitrogen metabolism by LAB [12,38]. Proteolytic activity of LAB is related to acid production and growth ability. Generally, LAB need free amino acids for growth. However, milk and dairy products, such as yogurt and cheese, have large-sized peptide and casein. LAB can break down casein to small-sized peptides and amino acids by their self-proteolytic system for growth [39].

Thus, *L. curvatus* in cheese-*ddukbokki* rice cake is expected to affect the growth of LAB due to its relatively high protease activity. As a result, casein of cheese is degraded to small molecules such as amino acids and peptides to be easily used. Thus, *L. curvatus* grows rapidly and produces relatively more organic acids. As a result, the pH value of cheese is decreased by *L. curvatus* and cheese-hardening of cheese-*ddukbokki* rice cake can rapidly occur at refrigerated storage.

## 4. Conclusions

In this study, various LAB strains were isolated from cheese-*ddukbokki* rice cake samples with rapid hardening. Their correlations with hardening of cheese were investigated. Among isolated LAB, *L. curvatus* was identified as the cause of cheese-hardening. Thus, we prepared cheese in which *L. curvatus* (LC-CD) or *L.*
*mesenteroides* (LM-CD, non-hardening strain) was the most dominant. Each *cheese-ddukbokki* rice cake made of prepared cheese was subjected to quality test during 50 days at 10 °C. As a result, *L. curvatus* caused hardening of cheese faster than other LAB strains due to its high protease activity and strong production of acid. Therefore, to improve the hardening of cheese, it is essential to control (inhibit) the growth of *L. curvatus*. Currently we are investigating whether we could directly or indirectly utilize of LAB having antimicrobial activity against *L. curvatus*.

## Figures and Tables

**Figure 1 microorganisms-09-01044-f001:**
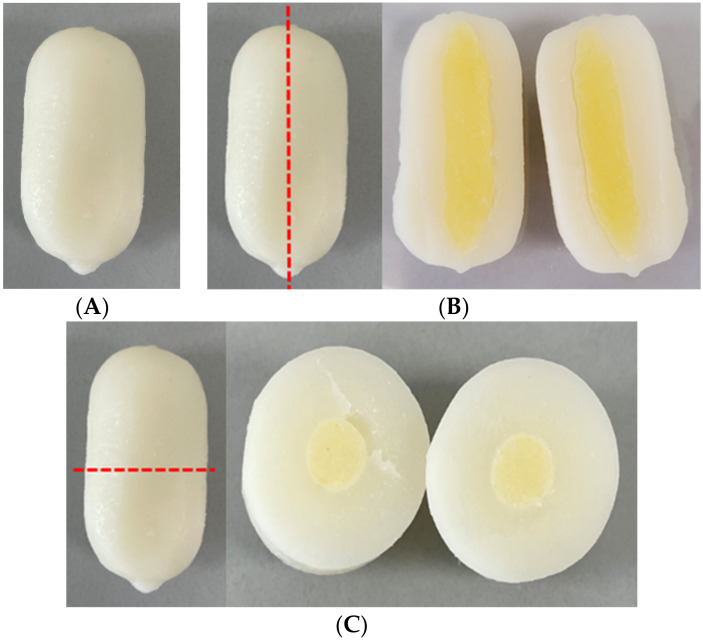
Structure of cheese-*ddukbokki* rice cake: (**A**) Whole product shape, (**B**) vertical section, (**C**) horizontal section.

**Figure 2 microorganisms-09-01044-f002:**
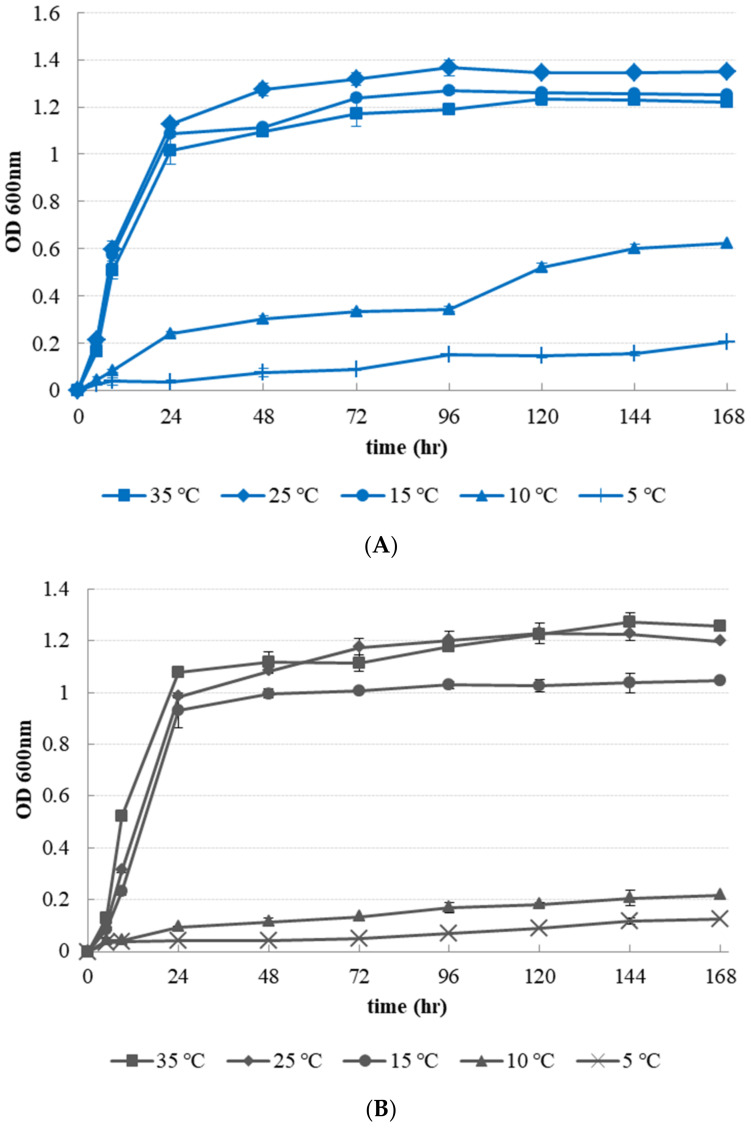
Growth of *Latilactobacillus curvatus* PD1 and *Leuconostoc*
*mesenteroides* PD2 at different temperature conditions in MRS broth with lactose (2%, *w/v*) for 7 days: (**A**) *L. curvatus* PD1 (**B**) *L.*
*mesenteroides* PD2.

**Figure 3 microorganisms-09-01044-f003:**
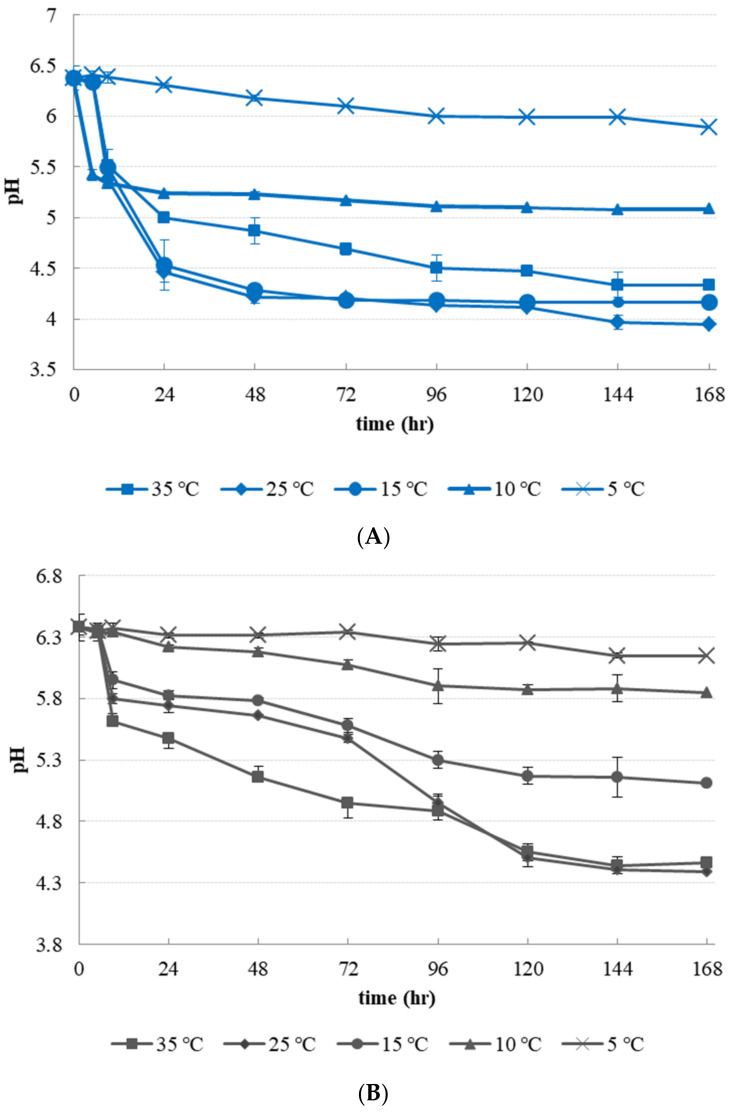
Change of pH values of *Latilactobacillus curvatus* PD1 and *Leuconostoc*
*mesenteroides* PD2 at different temperature conditions in MRS broth with lactose (2%, *w/v*) for 7 days: (**A**) *Lactobacillus curvatus* PD1 (**B**) *Leuconostoc*
*mesenteroides* PD2.

**Figure 4 microorganisms-09-01044-f004:**
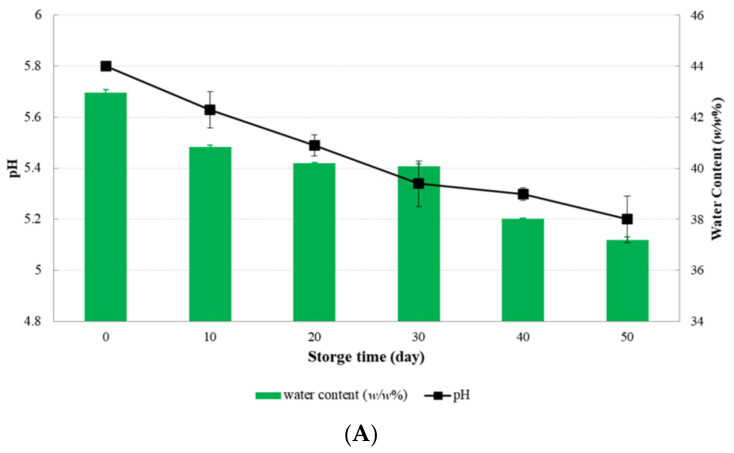
pH and Water content (%) of cheese-*ddukbokki* rice cake samples during 50 days at 10 °C: (**A**) Con-CD, (**B**) LC-CD, (**C**) LM-CD.

**Figure 5 microorganisms-09-01044-f005:**
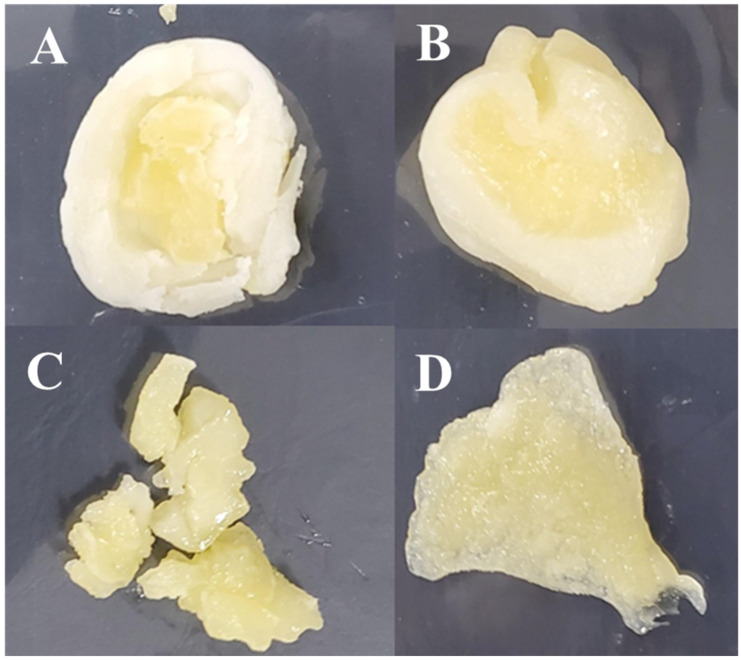
Meltability test of sliced LC-CD and LM-CD samples at storage times of 30 days: (**A**) Sliced LC-CD samples (**B**) sliced LM-CD samples (**C**), Cheese in which *Lactobacillus curvatus* was dominant (**D**) Cheese in which *Leuconostoc*
*mesenteroides* was dominant.

**Figure 6 microorganisms-09-01044-f006:**
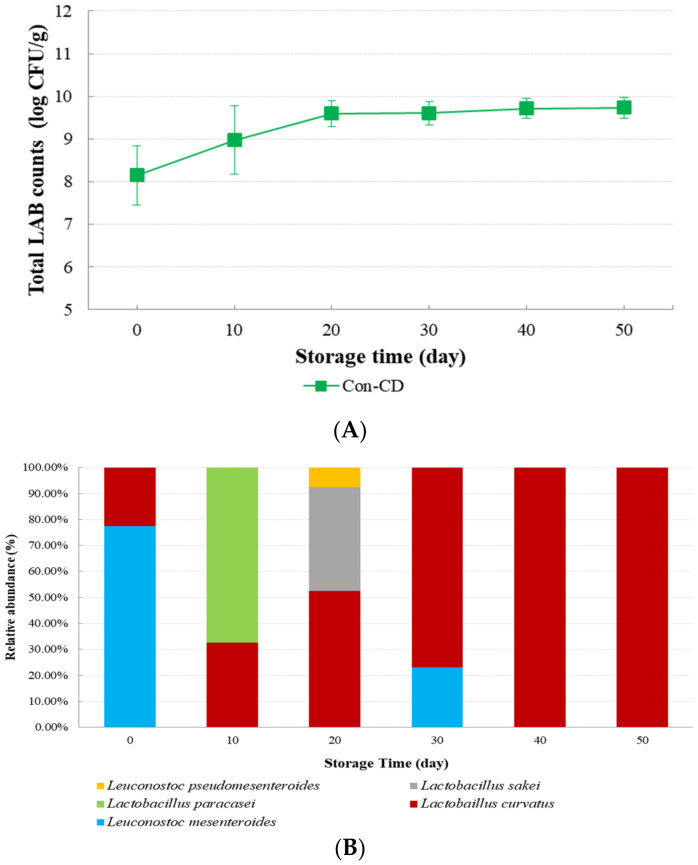
Total lactic acid bacteria cell counts and microbial community of Con-CD samples during 50 day at 10 °C: (**A**) Total LAB counts; (**B**) microbial community.

**Figure 7 microorganisms-09-01044-f007:**
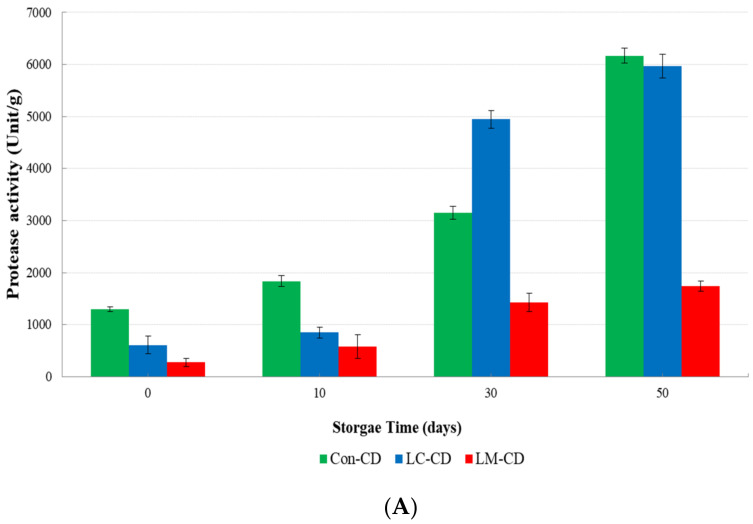
Proteolytic activites of LAB isolates and cheese-*ddukbokki* rice cake samples: (**A**) Cheese-*ddukbokki* rice cake samples at 10 °C for 50 days; (**B**) six LAB strains isolated from cheese-*ddukbokki* rice cake.

**Table 1 microorganisms-09-01044-t001:** Hardening properties and Identification of LAB from cheese-*ddukbokki* rice cake samples with different manufacturing dates.

Sample	Cheese’s Expired Dates	Identification of LAB	Colony Numbers out of 30	Hardening of Cheese	Sour Strength
*Cheese-ddukbokki*	13 November 2020	*Latilactobacillus curvatus*	30 (100%)	Strong	Strong
	13 November 2020	*Latilactobacillus curvatus*	30 (100%)	Strong	Strong
	21 November 2020	*Lacticaseibacillus paracasei* *Lacticaseibacillus casei* *Latilactobacillus curvatus*	9 (30.00%) 9 (30.00%) 12 (40.00%)	Weak	Weak
	21 November 2020	*Leuconostoc* *mesenteroides* *Lacticaseibacillus casei*	22 (73.33%) 8 (26.67%)	Non-hardening	None
	6 November 2020	*Leuconostoc mesenteroides*	30 (100%)	Non-hardening	None
	26 November 2020	*Leuconostoc mesenteroides*	30 (100%)	Non-hardening	None
	2 December 2020	*Leuconostoc mesenteroides*	30 (100%)	Non-hardening	None

**Table 2 microorganisms-09-01044-t002:** Identification of LAB from Pizza Shredded Ⅱ mix-5 cheese samples with different manufacturing dates.

Sample	Manufacture Dates	Identification of LAB	Colony Numbers out of 30
Pizza Shredded Ⅱ mix-5 cheese	20.08.14	*Latilactobacillus curvatus*	15 (50.00%)
		*Leuconostoc pseudomesenteroides*	9 (30.00%)
		*Lacticaseibacillus paracasei*	3 (10.00%)
		*Lacticaseibacillus casei*	3 (10.00%)
	20.08.22	*Latilactobacillus curvatus*	30 (100%)
	20.08.27	*Latilactobacillus curvatus*	30 (100%)
	20.09.03	*Leuconostoc mesenteroides*	30 (100%)
	20.09.18	*Leuconostoc mesenteroides*	21 (70.00%)
		*Latilactobacillus curvatus*	9 (30.00%)
	20.09.24	*Leuconostoc mesenteroides*	18 (60.00%)
		*Latilactobacillus curvatus*	12 (40.00%)
	20.10.02	*Latilactobacillus curvatus*	30 (100%)
	20.10.08	*Leuconostoc mesenteroides*	21 (70.00%)
		*Latilactobacillus curvatus*	9 (30.00%)
	20.10.15	*Latilactobacillus curvatus*	15 (50.00%)
		*Latilactobacillus sakei*	15 (50.00%)
	20.10.26	*Latilactobacillus curvatus*	30 (100%)
	20.11.05	*Leuconostoc mesenteroides*	12 (40.00%)
		*Levilactobacillus brevis*	6 (20.00%)
		*Lacticaseibacillus casei*	12 (40.00%)
	20.12.11	*Leuconostoc mesenteroides*	15 (50%)
		*Lacticaseibacillus casei*	15 (50%)

**Table 3 microorganisms-09-01044-t003:** Meltability test of cheese-*ddukbokki* rice cake samples during 50 days at 10 °C.

Storage Time (Day)	Con-CD ^1^	LC-CD ^2^	LM-CD ^3^
0	Melted	Melted	Melted
10	Melted	Melted	Melt
20	Melted	Melted	Melted
30	Melted	Non-melting	Melted
40	Melted	Non-melting	Melted
50	Non-melting	Non-melting	Melted

^1^ Con-CD, Commercially available CD samples containing cheese with unknown dominant strains. ^2^ LC-CD. Manufacturing CD samples containing cheese with the dominant strain being *Latilactobacillus curvatus*. ^3^ LM-CD. Manufacturing CD samples containing cheese with dominant strain being *Leuconostoc mesenteroides*.

**Table 4 microorganisms-09-01044-t004:** Microbial community of lactic acid bacteria of LC-CD and LM-CD during 30 days at 10 °C.

Sample	Storage Time(Day)	Identification of LAB	Colony Numbers out of 30 (%)
LC-CD	0	*Lactobacillus curvatus*	30/30 (100%)
	30	*Lactobacillus curvatus*	30/30 (100%)
LM-CD	0	*Leuconostoc mesenteroides*	30/30 (100%)
	30	*Leuconostoc* *mesenteroides* *Lactobacillus curvatus*	24/30 (80%) 6/30 (20%)

## Data Availability

Not applicable.

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
