# Peer review of "Hardening Properties of Cheeses by Latilactobacillus curvatus PD1 Isolated from Hardened Cheese-Ddukbokki Rice Cake"

_microorganisms, 2021, doi:10.3390/microorganisms9051044_

Round 1

Reviewer 1 Report

The correlation of LAB strains isolated from Cheese-dukbokki rice cake (CD) samples with cheese hardening was investigated in this study. The species associated with a faster hardening of the cheese was L. curvatus. After various tests for 50 days it was found that this was related to the higher protease activity and acid production. Therefore, it would be necessary to inhibit the multiplication of L. curvatus for the proper hardening of the cheese. The study is sound, clear and well written, but I would like to point out a few doubts below:

- The authors selected 30 colonies at random after cultivating serial dilutions. It is not clear whether these colonies were selected from the same Petri dish (or dilution). Please clarify this doubt.
- Also, what was the identity percentage in BLAST for LAB isolates?

Minor comments:

- Line 79: Mikelsone should not be in italics;

- Section 2.1, page 3, please add the reference for primers 27F and 1492R;

- Line 191: L. lactis should be in italics;

- Line 234: genera is misspelled (gena); “When” should not be in italics;

- Table 2, page 10, please, authors should add explanatory footnotes to the abbreviations used in the table;

Author Response

Thank you very much for your excellent review report. We responded to your questions , and the manuscript was modified accordingly. The response report to the review report was uploaded as a file. 

Reviewer 2 Report

Please refer to my comments in the attached file. 

Thank you.

JS

Author Response

Thank you very much for your excellent review report. We responded to your advises , and the manuscript was modified accordingly. The response report to the review report was uploaded as a file. 

Round 2

Reviewer 2 Report

Dear authors,

Please refer to the attached file with my 2nd round review comments. I think that now minor revisuions are needed, but please consider more carefully my observations and advice regarding the quality of your revision process overall. Thank you 

Author Response

First of all, thank you for your great and detailed advise. We are very sorry that the correction and answer for the 1st revision were not made completely. We uploaded answers and revisions for the 2nd revision report in today. The modified sentences in the manuscript are marked in green color. The revision reports were written and attached as a word file.
